# Mesenchymal Stromal Cells for the Enhancement of Surgical Flexor Tendon Repair in Animal Models: A Systematic Review and Meta-Analysis

**DOI:** 10.3390/bioengineering11070656

**Published:** 2024-06-27

**Authors:** Ilias Ektor Epanomeritakis, Andreas Eleftheriou, Anna Economou, Victor Lu, Wasim Khan

**Affiliations:** 1Department of Surgery, Addenbrooke’s Hospital, University of Cambridge, Cambridge CB2 0QQ, UK; iee21@cantab.ac.uk; 2School of Clinical Medicine, University of Cambridge, Cambridge CB2 0SP, UK; eleftheriou.a@outlook.com (A.E.); ae430@cam.ac.uk (A.E.); victorluwawa@yahoo.com.hk (V.L.); 3Department of Trauma and Orthopaedic Surgery, Addenbrooke’s Hospital, University of Cambridge, Cambridge CB2 0QQ, UK

**Keywords:** flexor tendon, biomechanics, mesenchymal stromal cells, repair and regeneration

## Abstract

Flexor tendon lacerations are primarily treated by surgical repair. Limited intrinsic healing ability means the repair site can remain weak. Furthermore, adhesion formation may reduce range of motion post-operatively. Mesenchymal stromal cells (MSCs) have been trialled for repair and regeneration of multiple musculoskeletal structures. Our goal was to determine the efficacy of MSCs in enhancing the biomechanical properties of surgically repaired flexor tendons. A PRISMA systematic review was conducted using four databases (PubMed, Ovid, Web of Science, and CINAHL) to identify studies using MSCs to augment surgical repair of flexor tendon injuries in animals compared to surgical repair alone. Nine studies were included, which investigated either bone marrow- or adipose-derived MSCs. Results of biomechanical testing were extracted and meta-analyses were performed regarding the maximum load, friction and properties relating to viscoelastic behaviour. There was no significant difference in maximum load at final follow-up. However, friction, a surrogate measure of adhesions, was significantly reduced following the application of MSCs (*p* = 0.04). Other properties showed variable results and dissipation of the therapeutic benefits of MSCs over time. In conclusion, MSCs reduce adhesion formation following tendon injury. This may result from their immunomodulatory function, dampening the inflammatory response. However, this may come at the cost of favourable healing which will restore the tendon’s viscoelastic properties. The short duration of some improvements may reflect MSCs’ limited survival or poor retention. Further investigation is needed to clarify the effect of MSC therapy and optimise its duration of action.

## 1. Introduction

Injuries to the hand and wrist are common, accounting for up to 20% of presentations to the Emergency Department [1,2,3].These range from uncomplicated, self-resolving injuries to a cause of long-term disability, incurring substantial costs to healthcare systems, individuals and society [1]. Flexor tendon lacerations may occur in isolation or in conjunction with more complex hand injuries [4]. One study demonstrated that the cost to American healthcare systems resulting from flexor tendon lacerations alone was greater than USD 240 million per year [5]. Days missed from work accounted for most of the resulting societal costs. As well as the functional consequences of such injuries, post-traumatic stress disorder related to the workplace may complicate an individuals’ return to work [6]. Optimization of treatment of these injuries, therefore, represents an important focus for restoring quality of life after traumatic injury. 

Flexor tendon injuries may be classified anatomically into zones [7]. Regardless of the zone of injury, primary repair using suture material is the preferred method for repairing flexor tendon lacerations [8]. Restoration of function requires precise apposition of the tendon stumps to confer strength to the repaired tendon. This is commonly followed by early post-operative mobilization which aims to prevent adhesion formation while avoiding rupture at the repair site [9,10]. Failure of the repair and adhesion formation are the two main post-operative complications [11]. Preventing these remains a priority for improving functional outcomes. 

Tendons are specialized connective tissues. They consist of a dense extracellular matrix (ECM), composed primarily of collagen type I, which is laid down by tenocytes, a fibroblast-like cell population responsible for ECM production and tissue homeostasis [12,13]. Collagen fibres are arranged in parallel along the tendon’s longitudinal axis and are, in turn, organized into a hierarchical fibrillar arrangement [12,14]. This arrangement confers the tensile strength required to transmit the forces of muscle contraction to bone, permitting movement. Additional ECM components include water and proteoglycans, which act to resist compression, and glycoproteins such as lubricin which reduces gliding friction between fascicles [15]. As well as their tensile strength, tendons demonstrate viscoelastic behaviour, meaning their stiffness increases as larger loads are applied [13,16]. This permits more effective transmission of forces from muscular contraction. Tendons of the hand are also enclosed within a synovial sheath which reduces friction when gliding over joint surfaces [14]. 

Following traumatic injury, the capacity of tendons to fully repair is limited due to their hypocellularity and limited vascularity [13,17]. The response of tendon tissue to acute injury consists of inflammatory, proliferative and remodelling repair processes. Repair ultimately results in scar tissue at the injury site, adhesion formation and a failure to restore the tendon’s original biomechanical properties [12,13,18]. In the case of intrasynovial flexor tendons, the healing process is complicated by the fact that the overlying synovial tissue also participates in repair, resulting in adherence of the tendon to its synovial sheath and restricted motion [19]. Therefore, while healing within the tendon tissue itself (intrinsic healing) is necessary for the restoration of tendon strength, the repair process of the overlying tissues (extrinsic healing) contributes to adhesion formation [14]. A key challenge for strategies aimed at improving functional outcomes is to mitigate adhesion formation while improving, or without compromising, the tensile strength of the repair site [19].

Several treatments, both surgical and non-surgical, have been investigated to improve tendon healing. Attempts utilizing various suture repair methods have been made to balance the mechanical strength of the suture with adhesion creation to improve tendon healing [20,21,22]. This includes suture patterns which avoid a bulky repair site in order to facilitate gliding [19]. When tested on animal models, non-surgical therapies like platelet-rich plasma (PRP) injections have shown promising results in facilitating healing of ruptured extra-synovial tendons [23,24]. Lubricating biomolecules, such as hyaluronic acid and lubricin, have shown success in reducing adhesion formation but may compromise the tendon’s tensile strength [25,26]. 

Tissue engineering techniques have become a popular focus of investigation for the future treatment of musculoskeletal disease, including tendon rupture [27,28]. Mesenchymal stromal cells (MSCs), also termed mesenchymal stem cells, routinely demonstrate benefit in randomized trials for the treatment of osteoarthritis [29,30]. They may also be promising in augmenting healing of rotator cuff tendinopathy and Achilles’ tendon injuries [31,32]. MSCs have been derived from multiple adult tissues, including the bone marrow, adipose tissue, peripheral blood, synovium, dental pulp, placenta, and umbilical cord [33]. They have a specific definition, imposed by the International Society for Cell Therapy [34]. This relates to their growth characteristics in 2D culture, expression of a specific combination of surface markers and their ability to differentiate into bone-forming cells.

The regenerative capacity and low immunogenicity of MSCs has attracted much attention for their potential use in augmenting healing and repair of musculoskeletal tissues [35]. They may also be combined with polymer scaffolds and biomolecules, such as growth factors, to further potentiate their regenerative function [13,36]. As well as evading immune responses due to their lack of self-antigens, they have been recognised to modulate immune responses to achieve an anti-inflammatory phenotype [37]. This may partially account for their efficacy in the treatment of pathological inflammation in vivo, in conjunction with the differentiation into repair tissues. 

As previously mentioned, healing of tendon tissue following rupture is characterized by the processes of inflammation, repair, and remodelling, which are unable to achieve complete regeneration [13]. Consequently, there have been attempts to harness the immunomodulatory and regenerative properties of MSCs to augment surgical repair of flexor tendons in vivo. This systematic review aims to delineate the effect of MSCs on the biomechanical properties of surgically repaired flexor tendons in animal models. 

## 2. Materials and Methods

This review was conducted in accordance with the Preferred Reporting Items for Systematic Reviews and Meta-Analyses (PRISMA) guidelines [38]. The completed PRISMA checklist can be found in Appendix A. A pre-defined protocol for completion of the review was registered in the International Prospective Register of Systematic Reviews PROSPERO (CRD42023394908). This review aimed to answer the following focused question: 

Does supplementing surgical repair of transected flexor tendons with mesenchymal stromal cells enhance the biomechanical properties of repaired tendons in animal models?

### 2.1. Search Algorithm

A systematic literature search was conducted from conception until February 2024 using the following databases: (1) PubMed, (2) OVID, (3) Web of Science, and (4) CINAHL. The detailed search strategy can be found in Appendix A. Studies obtained using the search strategy were uploaded onto the Rayyan website for screening [39]. Screening for inclusion was first performed by title and abstract using pre-determined inclusion and exclusion criteria, defined below. This was performed independently by AEl and AEc in a blinded fashion. Following unblinding, IEE was consulted to resolve disagreements in screening decisions. This was followed by screening of full-text articles using the same decision-making process.

### 2.2. Inclusion and Exclusion Criteria

Using the Population, Intervention, Comparison, Outcome, Study type (PICOS) model [40] as a guide, we formulated our inclusion and exclusion criteria for study selection (Table 1).

### 2.3. Data Extraction

Extraction of relevant data from the finally included full-text articles was performed by IEE, AEc and AEl. Tables created using Excel version 16.66.1 were populated with data pertaining to the parameters below. 

Study characteristics, including the study design, animal model, cohort size, tendon defect location, post-operative weight-bearing protocol and timing of sacrifice.The type of intervention, including MSC source, cell delivery method, composition of the delivery method, cell number and/or density, and method of surgical repair.Biomechanical properties including maximum load, surrogate measures of adhesion formation, maximum stress, maximum strain, elastic modulus, and energy absorption.

Relative to the pre-defined protocol, some amendments were made to data collection to permit focused analysis of the most clinically relevant biomechanical properties. However, this was performed without alteration of the inclusion or exclusion criteria or method of analysis. 

### 2.4. Data Analysis

Meta-analyses were conducted by VL for quantitative results of biomechanical tests which were deemed comparable. Comparable parameters were those tested using a similar experimental design and mechanical testing apparatus, resulting in the same units of measurement. For continuous data, the standardized mean difference (SMD) was used to pool the data and was reported together with the 95% confidence interval (CI) and *p*-value for comparisons between control and intervention groups. To account for differences between species, the SMD, or effect size, was used rather than raw data. 

Meta-analyses were carried out using RStudio version 4.0.5. for continuous data. The analytic code for performing meta-analyses and subgroup meta-analyses can be found in the Appendix A. The estimator reported by Hozo et al. was used where the standard deviation was not provided in the manuscript [41,42]. Higgins and Thompson’s I^2^ statistic and Cochran’s Q test were used as measures of heterogeneity [43,44]. Prediction intervals were also included. Follow-up data were grouped into two main timepoints to facilitate the meta-analysis: three weeks and eight weeks. Where two follow-up timepoints could be rounded up into the eight-week period, the time of final follow-up was used. Egger’s regression test was used to assess for publication bias and can be found in the Appendix A.

### 2.5. Assessing Risk of Bias

The risk of bias (RoB) arising from the included studies was assessed using the RoB 2.0 criteria, devised by Sterne et al. (2019) [45]. These criteria assess randomized trials according to five domains which are described below:
Bias arising from the randomization process
Was the allocation system random?Was the allocation sequence concealed until participants were enrolled and assigned to interventions?Did baseline differences between intervention groups suggest a problem with the randomisation process?
Bias due to deviations from the intended interventions
Were participants aware of their assigned intervention during the trial?Were carers and people delivering the interventions aware of participants’ assigned intervention during the trial?If Y/PY/NI to 2.1 or 2.2: Were there deviations from the intended intervention that arose because of the trial context?If Y/PY/NI to 2.3: Were these deviations likely to have affected the outcome?If Y/PY to 2.4: Were these deviations from intended intervention balanced between groups?Was an appropriate analysis used to estimate the effect of assignment to intervention?If N/PN/NI to 2.6: Was there potential for a substantial impact (on the result) of the failure to analyse participants in the group to which they were randomised?
Bias due to missing outcome data
Were data for this outcome available for all, or nearly all, participants randomised?If N/PN/NI to 3.1: Is there evidence that the result was not biased by missing outcome data?If N/PN to 3.2: Could missingness in the outcome depend on its true value?If Y/PY/NI to 3.3: Is it likely that missingness in the outcome depended on its true value?
Bias in measurement of the outcome
Was the method of measuring the outcome inappropriate?Could measurement or ascertainment of the outcome have differed between intervention groups?If N/PN/NI to 4.1 and 4.2: Were outcome assessors aware of the intervention received by study participants?If Y/PY/NI to 4.3: Could assessment of the outcome have been influenced by knowledge of intervention received?If Y/PY/NI to 4.4: Is it likely that assessment of the outcome was influenced by knowledge of intervention received?
Bias in selection of the reported result
1.Were the data that produced this result analysed in accordance with a prespecified analysis plan that was finalised before unblinded outcome data were available for analysis?   Is the numerical result being assessed likely to have been selected, on the basis of the results, from the following:2.Multiple eligible outcome measurements (e.g., scales, definitions, time points) within the outcome domain?3.Multiple eligible analyses of the data?


These domains were each assessed as to whether the included studies demonstrated low risk, some concerns, or high risk of bias, and an overall risk was determined. Results of the assessments are presented graphically using the robvis package in RStudio [46]. IEE and VL independently carried out the risk of bias assessments. There were no disagreements in the outcomes of these assessments. 

## 3. Results

### 3.1. Search Results

The structured search, using five databases, yielded 2830 papers in total (Figure 1). After removal of duplicates, 1834 articles remained, of which 1807 were excluded following title and abstract screening. Of the remaining 27 studies which underwent full-text screening, 9 were eligible for inclusion [47,48,49,50,51,52,53,54,55]. 

### 3.2. Study Characteristics

Table 2 presents a summary of individual study characteristics, including animal subject characteristics, the interventions studied, post-operative weight-bearing and timing of sacrifice. The nine included studies were randomized controlled trials (RCTs), investigating the effect of MSCs on flexor tendon repair in comparison to cell-free controls and suture repair alone. The animal subjects were either New Zealand white rabbits [47,48,49,51,54,55] or dogs [50,52,53]. Eight studies [47,48,49,50,51,52,53,54] used transection of the flexor digitorum profundus (FDP) as their injury model and one used the flexor digitorum superficialis (FDS) tendon [55]. Four studies investigated adipose-derived MSCs [47,48,53,55] and bone marrow-derived MSCs [50,51,52,54] in isolation, while one compared the two tissue sources [49]. The cell dosage per treatment varied from 10^5^ to 4 × 10^6^ cells. Intratendinous injection was the most commonly employed cell delivery method [46,47,48,54], followed by pipetted droplets [50,51,52] and MSCs embedded in scaffolds [53,54]. The composition of the therapy varied, and included phosphate buffered saline (PBS) [47,48,49], collagen [50,52], hyaluronic acid [50,52,54], L-lactide and ℇ-caprolactone (PLCL) [54], and fibrin [51,53]. Two studies supplemented the cell therapy with growth differentiation factor 5 (GDF-5) and the lubricating glycoprotein lubricin [50,52] and another used bone morphogenetic protein 12 [53]. Following surgery, five studies employed immobilization in the immediate post-operative period [47,48,49,50,52]. Four performed procedures to unload the healing tendon in addition to the tendon transection itself [50,51,52,54]. Three studies permitted free movement immediately after surgery [51,54,55] while another involved controlled passive motion exercises until the time of sacrifice [53]. The timing of sacrifice spanned 10 days to 8 weeks post-operatively. 

### 3.3. Mechanical Properties

The results of mechanical testing are shown in Table 3. 

#### 3.3.1. Maximum Load

Maximum load refers to the greatest force, in Newtons, which can be applied across the tendon before rupture occurs. All of the included studies reported this parameter in a comparable manner. Results available from three- and eight-week timepoints were pooled in meta-analyses (Figure 2). These revealed no significant change in the maximum load of tendons when surgical repair was supplemented with MSCs.

The studies demonstrated a variable effect of MSC supplementation on the maximum load. At final follow-up, five studies showed a significant improvement [47,48,49,54,55], of which four used adipose-derived MSCs [47,48,49,55]. Three studies using bone marrow-derived MSCs showed a significant reduction in maximum load [50,51,52]. The overall result was a non-significant SMD of 0.40 (95% confidence interval (CI) (−0.53, 1.33), *p* = 0.36). However, this did show a relative improvement compared to three-week follow-up (SMD: 0.18, 95% CI (−1.40, 1.77), *p* = 0.80).

#### 3.3.2. Adhesions

Multiple surrogate measures of adhesion formation were used across the included studies. These included work of flexion (N/mm/degree), gliding resistance or friction (N) and range of motion (ROM) of the proximal and distal interphalangeal joints in degrees. Friction was consistently reported in two studies [50,52] and results were analysed in meta-analyses at three- and six-week timepoints (Figure 3). Although the initial result was not significant (SMD: −1.03, 95% CI (−3.55, 1.50), *p* = 0.12), at six weeks a statistically significant reduction in friction was noted relative to suture repair alone (SMD: −3.30; 95% CI (−6.10, −0.50), *p* = 0.04). These studies, performed by Zhao and colleagues [50,52], involved applying BMSCs suspended in a collagen gel and supplemented with GDF5. Notably, this was followed by additional treatment with a lubricating hyaluronic acid–lubricin composite. There was no comparison of MSC treatments with and without this additional lubricant or of the lubricant in isolation.

Similar trends were noticed regarding the work of flexion in the same studies. Range-of-motion analyses demonstrated varied results. While, at three weeks, He et al. (2015) showed a significant improvement in ROM following pipetting of 4 × 10^6^ around the repair site [51], this was not maintained at final follow-up. Gelberman et al. (2016) used a heparin/fibrin-based scaffold implanted within the tendon itself at the time of repair. This resulted in a significantly reduced ROM relative to controls, which the authors suggest may be due to a physical hindrance caused by the scaffold [53]. This is supported by the fact that no significant difference in adhesion formation was noted between groups when examined macroscopically.

#### 3.3.3. Viscoelastic Properties

Stress, measured in pascals, is calculated by dividing the force applied to the tendon by its cross-sectional area. A meta-analysis of results at three weeks demonstrated a significant increase in the maximum stress (SMD: 2.02; 95% CI (0.40, 3.65), *p* = 0.02) (Figure 4). However, this effect was not maintained at eight weeks (SMD: 0.63; 95% CI (−2.13 to 3.39), *p* = 0.61). The most notable improvement was seen in the studies conducted by Behfar and colleagues [47,48,49], which involved intra-tendinous injections of either adipose-derived stromal vascular fraction or bone marrow-derived MSCs in PBS. He et al. (2015) tested the effect of varying doses of allogenic or autologous bone marrow-derived MSCs suspended in a glue composed primarily of fibrinogen and thrombin [51]. This was pipetted around the repair site. At eight weeks, all permutations of this intervention revealed declines in the maximum stress.

Strain represents the degree of deformation of the tendon when force is applied and is expressed as a percentage. Three studies reported this outcome measure [48,49,53]. The pooled analysis for maximum strain is shown in Figure 5. This demonstrated a decline in the maximum strain relative to control tendons undergoing suture repair alone at both three (SMD: −2.13; 95% CI (−4.18, −0.09), *p* = 0.05) and eight (SMD: −1.35; 95% CI (−4.04, 1.34), *p* = 0.21) weeks. However, this was only significant at three-week follow-up.

The Young’s modulus, or elastic modulus, is calculated by dividing the stress of a material by its strain. This equates to the stiffness of the material, or its capacity to resist deformation when force is applied. Two studies were included in the meta-analysis of the Young’s modulus (Figure 6) [51,55]. This showed a decline at the time of final follow-up relative to suture repair alone (SMD: −1.26; 95%; CI (−2.62, 0.11), *p* = 0.06).

Energy absorption is a measure of the compliance of a material when force is applied. Four studies were eligible for inclusion in the meta-analysis of energy absorption (Figure 7) [47,48,49,53]. At the time of final follow-up, the energy absorption demonstrated a statistically significant increase relative to suture repair alone (SMD: 2.06; 95% CI (1.11, 3.01), *p* = 0.004). All of the studies included in this analysis administered adipose-derived MSCs, three by intra-tendinous injection and one using cells embedded in a heparin/fibrin-based scaffold [53].

### 3.4. Subgroup Meta-Analyses

Subgroup-analyses were conducted to determine the differential impact of the MSC source on tendon healing, distinguishing between adipose-derived and bone marrow-derived MSCs at three- and eight-weeks post-treatment (Table 4). Each MSC source was assessed in isolation by four studies and one study compared the two sources. The latter study, conducted by Behfar et al. (2013) found that adipose-derived stromal vascular fraction resulted in a statistically significant increase energy absorption and reduced maximum stress [49].

Of the mechanical properties assessed in this review, only the maximum load and stress were suitable for subgroup meta-analysis according to MSC source. At three weeks, adipose-derived MSCs were associated with a decrease in maximum load relative to bone-marrow MSCs (*p* = 0.78). However, by eight weeks, adipose-derived MSCs exhibited a positive shift, exceeding the performance of bone marrow-derived MSCs (*p* = 0.07). Although the difference between the two groups was not statistically significant at either timepoint, the results suggest variable therapeutic efficacy based on MSC origin and the timing of final follow-up.

Subgroup analysis of the maximum stress demonstrated an improvement following treatment with adipose derived MSCs at both timepoints, although this declined somewhat by the time of eight-week follow-up. In contrast, the use of bone marrow MSCs demonstrated an initial improvement at three weeks but this effect deteriorated by eight weeks. Again, the difference between the groups was not statistically significant at either timepoint but does demonstrate a differential impact of the MSC tissue source and the potential impact of follow-up time on tendon healing.

### 3.5. Risk of Bias Assessment

Risk of bias assessment demonstrated an overall high risk of bias (Figure 8). Concerns arose primarily from the randomization process, as studies did not conceal allocation to a particular treatment and baseline characteristics of different groups were not specified. Furthermore, the risk of bias was high regarding measurement of outcomes, as it was unclear whether knowledge of the intervention received will have influenced measurements. The outcome of RoB assessments for each of the included studies can be found in Appendix A.

## 4. Discussion

This review aimed to assess the effect of augmenting the surgical repair of flexor tendons using MSCs. The supplementation of surgical repair with tissue engineering techniques is aimed at mitigating the two main post-operative complications following tendon repair in the clinical setting: rupture at the repair site and adhesion formation. While MSCs have been trialled in the context of various musculoskeletal diseases, their use for tendon repair remains a relatively novel area of investigation. To our knowledge, only pre-clinical trials have been performed.

This paper included nine studies presenting quantitative data regarding mechanical properties of treated tendons. Meta-analyses demonstrated reduced friction following the use of MSCs, representing reduced adhesion formation. The maximum load was not compromised. However, the viscoelastic properties of treated tendons were altered such that the elastic modulus was reduced and energy absorption increased. Clinically, these findings may manifest as decreased hand stiffness following surgery, but reduced efficiency of energy transfer between muscle and bone. These concepts and the possible underlying mechanisms are explored below. However, the interpretation of these findings is obscured by several limitations, such as the heterogeneity of included studies and limitations inherent in pre-clinical trials. Below, we discuss the interpretation of these findings in more detail, considerations for future research, and the important limitations of this review.

### 4.1. Biomechanical Properties

#### 4.1.1. Maximum Load and Adhesion Formation

Biomechanical testing is typically performed by mounting tendons, harvested after sacrifice of the animal subjects, onto a testing apparatus [56]. A load is applied in predetermined increments across the repair site. The maximum load refers to the greatest force applied, in Newtons, until rupture occurs. The tensile strength and other mechanical properties of the repaired tendon are largely a function of collagen type I deposition after injury, and depend on the diameter, orientation and degree of crosslinking between deposited collagen type I fibres [16,51,57,58]. The initial tendon ECM after injury demonstrates an increase in randomly oriented collagen type III, seen macroscopically as granulation tissue, which is gradually replaced by type I collagen during remodelling [58]. Although collagen type III is present in the native tendon and serves to crosslink type I collagen fibres [16], its persistent abundance after injury is associated with scar tissue formation and suboptimal mechanical properties. This meta-analysis demonstrated that maximum load did not improve significantly, relative to suture repair alone. Of the studies in which a significant improvement was seen in maximum load, the majority involved the application of adipose-derived regenerative therapy. Although subgroup analysis did favour the use of adipose therapy, this difference was not significant.

This is contrasted with the results of surrogate measurements of adhesion formation, particularly friction. Zhao et al. [50,52] showed that, despite a significant decrease in maximum load, there was a significant reduction in friction following treatment with bone marrow-derived MSCs supplemented with a hyaluronic acid–lubricin composite and growth and differentiation factor 5 (GDF-5). This may be explained by the capability of MSCs to dampen inflammation and fibrosis through the secretion of soluble mediators and extracellular vesicles in response to pro-inflammatory stimuli [59,60]. In the context of tendon repair specifically, this is supported by data showing the ability of adipose-derived MSCs (ASCs) to reduce the expression of pro-inflammatory cytokines and fibrotic gene expression in a canine flexor tendon injury model [61]. However, the potential independent role of the hyaluronic acid–lubricin composite, which serves as a lubricant, cannot be discounted, and direct comparison between this and MSCs alone is necessary.

The reasons for the inconsistent, and sometimes conflicting, effect of MSC therapy on the maximum load of repaired tendons remain unclear. Both generally and in the context of tendon repair, MSCs have shown beneficial effects in various phases of wound healing by modulating the inflammatory response, promoting angiogenesis and fibroblast proliferation, and accelerating ECM remodelling [62,63]. In the case of intrasynovial flexor tendons, the distinction between intrinsic healing of the tendon itself and extrinsic healing of the surrounding tissue [64,65] may be of relevance. While intrinsic healing is necessary to restore tendon strength, extrinsic healing, which predominates in the early stages after injury, is primarily responsible for adhesion formation [64,65]. Zhao et al. [50] note that adhesion formation itself may be responsible for the increased maximum load observed in repaired tendons not treated with MSC therapy. It is possible that, within the limited timeframe of the included studies, the effect of MSCs on reducing the fibrotic healing of extrinsic tissues led to an unexpected initial decline in maximum load in some studies.

Heterogeneity between studies regarding the method of cell delivery might also explain the varied success of tendon healing. Studies involving administration of cells in droplet form tended to demonstrate a lower maximum load than those using intratendinous injection or MSCs embedded in scaffolds. Improving the retention of exogenously delivered MSCs is a widely recognised challenge in cell engineering therapy for multiple conditions [30,66,67]. This is partly explained by cell leakage, which may be addressed by embedding MSCs in polymer scaffolds [68]. This has, for example, demonstrated success in the field of MSC implantation therapy for cartilage regeneration [69]. Frauz et al. (2019) have shown the ability of a biodegradable fibrin-sealant scaffold to enhance retention of adipose-derived MSCs in a mouse model of tendon injury without a detrimental effect on tendon healing [70]. While the physical distribution of MSCs away from the injury site may explain the varying results obtained in this review, it is important to also recognise the need to optimise cell survival [66,67,71]. Cells delivered to an injured environment are subject to inflammation, hypoxia and oxidative stress. Pre-treatment of MSCs in hypoxic environments and genetic modification to enhance anti-apoptotic gene expression are strategies which have shown beneficial results for cell survival [66,67,71]. An additional strategy, which has been used for the delivery of adipose-derived MSCs, is to administer cells within their native matrix, as found in vivo [72]. This has proven beneficial for physical retention of MSCs within the graft and their protection from the injurious environment.

#### 4.1.2. Viscoelastic Behaviour

Tendons have a characteristic stress–strain curve during stretching [13,14]. The initial toe region is non-linear and represents the point at which collagen fibrils are crimped and therefore relatively easily stretched. This is followed by a linear portion, during which collagen fibres are straight and stress is directly proportional to strain. The gradient of this region is termed the Young’s modulus, which quantifies the tendon’s stiffness or resistance to elongation. Strain is a measure of the relative deformation of a material when a load is applied [73]. The maximum strain of tendons in vivo depends on their relative contribution to either positional maintenance or participation in locomotion and may vary between 2 and 10% under physiological conditions [15]. Beyond the maximum strain, mechanical failure of collagen fibres occurs, leading to macroscopic tears and tendon rupture [13,14].

The dependence of tendon stiffness on the rate of mechanical strain is termed viscoelasticity [16]. Viscoelastic behaviour dictates that at low strain rates the energy absorption capacity, or compliance, of the tendon is increased, permitting the tendon to deform to a greater degree. At higher strain the tendon is stiffer and deforms less. This has consequences for force transmission from muscle to bone [14,16]. Increased stiffness (i.e., a higher Young’s modulus) equates to greater efficiency of load transfer. However, lower energy absorption capacity may increase susceptibility to tendon rupture [74]. Therefore, while energy absorption must be sufficient to avoid permanent deformation of the tendon tissue, excessively low tendon stiffness will compromise transfer of forces to bone during muscle contraction.

The meta-analyses presented here demonstrate interesting consequences for the viscoelastic properties of treated tendons following the administration of MSC therapy. Although not statistically significant, the maximum strain was reduced in comparison to suture-only controls, meaning a lower percentage deformation was necessary to reach mechanical failure. The Young’s modulus was also reduced at final follow-up, equating to a lower gradient of the stress–strain curve and reduced resistance to stretching. Coinciding with this was a statistically significant increase in the energy absorption of tendons treated with MSCs at final follow-up, which represents a relative increase in the degree of stretching when a given force is applied. In vivo, this may manifest as an advantageous protection from subsequent re-injury but could also be interpreted as a compromised capacity for transfer of energy from muscle to bone. In vivo mechanical testing of tendons may be performed using ultrasound or magnetic resonance imaging (MRI) [75]. Studies employing these methods of post-operative mechanical evaluation would be valuable for the evidence base surrounding the use of MSCs to augment tendon repair.

As with the tensile strength, the viscoelastic behaviour of tendons varies according to ECM composition and organisation [57,76]. During postnatal development, the toe region of the stress–strain curve is elongated, representing increased compliance and, therefore, energy dissipation during tendon stretching [76]. The diameter, alignment and degree of cross-linking of collagen fibres alters during maturation, such that the tendons’ stiffness and Young’s modulus increase and the toe region shortens. This reflects reduced crimping and earlier recruitment of collagen fibres.

The random fibrillar arrangement of scar tissue might explain the increased energy absorption and decreased Young’s modulus observed following injury. Relating this to the findings of this review, it is possible that the administration of MSCs compromised ECM remodelling after suture repair, or that the follow-up time was insufficient to demonstrate the beneficial effect of MSC administration. Given the fact that MSCs promote tissue regeneration by modulating various aspects of the wound healing process, it is unclear why there should be a detrimental effect following their administration. MSCs have been shown to upregulate collagen gene expression by fibroblasts [28]. However, it is possible that the resulting fibrillar arrangement is disordered and not conducive to improved biomechanical properties. This is supported by one study in which treatment of transected mouse calcaneus tendons with an empty fibrin-sealant scaffold demonstrated superior ECM organisation in comparison to scaffolds embedded with adipose-derived MSCs three weeks after injury [70]. In contrast, another study demonstrated that at three weeks collagen type I deposition was denser and more organized following treatment of rabbit Achilles tendon with bone marrow-derived MSCs embedded in fibrin [77]. However, no differences were observed at later time points. Again, it is important to consider the limited follow-up time when interpreting these results.

### 4.2. Challenges in Tendon Tissue Engineering and Considerations for Future Research

#### 4.2.1. Mesenchymal Stromal Cells

MSCs have gained popularity as a cell source for tissue engineering, having repeatedly shown a capacity for promoting tissue regeneration [59,78]. This occurs through a combination of direct differentiation into specialized tissues, their immunomodulatory potential, and a protective paracrine effect on native cells. Their abundance and ease of isolation from multiple adult tissues of mesenchymal lineage, as well as their low immunogenicity, also make exogenously delivered MSCs an attractive therapeutic strategy [79,80]. Despite these advantages, there remain multiple barriers to the translation of MSC therapies into clinical settings, both generally and in relation to flexor tendon repair.

The potential application of such therapies for tendon repair in humans will depend on consistent demonstration of therapeutic benefit. To improve the quality of the available evidence base, standardization of the therapeutic approach is necessary. The results of the meta-analyses presented above frequently demonstrated statistically significant heterogeneity. This may have arisen due to variability in multiple aspects of the therapeutic strategy, including the MSC source, method of administration and whether augmentation with various tenogenic factors or lubricants was involved. The animal model used, surgical technique, and different protocols for post-operative weight-bearing will also have an impact.

The elucidation of whether adipose-derived or bone marrow-derived MSCs are superior in promoting tendon repair is one priority for standardizing therapy. Although BMSCs were the traditionally studied MSC source in tissue engineering, ASCs are growing in popularity due to their relative ease of isolation from adipose tissue and evidence that they possess greater proliferative and immunomodulatory potential [81,82,83]. Conversely, the tenogenic differentiation of BMSCs has been shown to be superior to that of ASCs, both in vivo and in vitro [84]. This raises important considerations for tailoring MSC therapy to the nature of the underlying pathology.

The degree of manipulation of the harvested tissue is also an important consideration for developing tissue engineering therapies. Behfar and colleagues used stromal vascular fraction [46,47,48], derived from the enzymatic digestion of adipose tissue. Relative to BMSCs, this resulted in significant differences for the biomechanical properties of treated tendons. Although SVF contains MSCs, it is also composed of various other cell components which may influence its behaviour when administered in vivo [85,86]. An additional potential strategy is to use mechanically disrupted adipose tissue [72]. This preserves the adipose ECM seen in vivo which improves retention of cells at the target site and affords protection from an inflammatory microenvironment [72].

An additional challenge facing the routine introduction of MSC therapies to clinical settings in general is safety concerns. There is a risk of allergic reactions to animal-derived components of culture media used for preparation of these therapies, as well as the theoretical potential for tumorigenesis, given that MSCs demonstrate increased genetic instability during prolonged culture [87]. MSC products have, however, repeatedly demonstrated an acceptable safety profile in human populations across many trials testing their efficacy for numerous indications [88].

It is also important to understand why, despite the promise of many pre-clinical studies, similar therapeutic benefits may not be demonstrated in human studies [88]. A potential explanation is the heterogeneity arising from the use of different tissue sources, in vitro preparation methods, and the difficulty in stratifying a heterogenous population of donors and recipients [78,82,88]. The method of administration of MSCs is also a significant source of variability among studies; the impact of various techniques on survival and retention of transplanted cells has already been mentioned. Furthermore, even MSCs obtained from the same tissue source can demonstrate heterogeneity, with consequences for their therapeutic efficacy [78]. Standardization of the various steps involved, from the harvesting of cells to the point of delivery, is therefore a key priority if consistent therapeutic efficacy is to be achieved. This includes accurate characterization of the cell population under investigation. As a minimum standard, this should include evidence that the MSC population meets the criteria set out by the International Society for Cellular Therapy (ISCT) to define MSCs [34]. This represents a significant limitation of the studies included in this review, as none of the included articles reported characterization of the cell populations identified as MSCs.

#### 4.2.2. Experimental Models

Animal models of tendon injury allow the detailed assessment of tissue properties after sacrifice. However, it is important to acknowledge the difficulties in accurately replicating the human phenotype of the disease and its recovery [89]. Flexor tendons of quadrupedal animals will naturally be exposed to different magnitudes of force and play a role in locomotion, unlike in humans. Attempting to model post-operative conditions on clinical practice would be of benefit to tailoring the experimental model. However, this is complicated by the fact that there remains ambiguity regarding the most appropriate post-operative rehabilitation protocol [90,91].

The Orthopaedic Research Society has recognised that there is no single animal model which is superior for studying the repair of a particular tendon [92]. For this reason, it is recommended that the animal used should be justified in the context of the injury under investigation. Murine flexor tendons are considered to have an anatomical resemblance to human flexor tendons due their surrounding synovial membrane. This is an important factor which should dictate tissue engineering strategies, given the synovial membrane’s impact on tissue healing and tendon gliding in vivo [51,65].

The papers included in this review utilized either rabbit or canine models of injury. Although there is reference to the suitability of each model for the study of human disease [47,53], this is not justified, for example, in terms of comparable anatomy. Additional consideration should be given to variations in post-operative weight-bearing protocols across the included studies. Avoidance of weight-bearing of the repaired tendon was achieved through various means, including cast immobilization [47,48,49], division of the tendon proximal to the site of transection [51,54], or by neurectomy to achieve paralysis of the limb extensors [50,52]. Other studies permitted free movement immediately after surgery [51,54,55], while another involved controlled passive motion exercises until sacrifice [53]. Given variations in post-operative rehabilitation practices, it can be argued that the experimental protocol employed should be justified in terms of intended clinical practice. Some authors state that avoidance of weight bearing does mirror certain clinical scenarios, such as immobilization in paediatric or uncooperative patients [52,54].

An additional consideration when designing models of flexor tendon injury relates to the presentation of these injuries in humans. Zone II flexor tendon injuries are commonly complicated by retraction of the proximal tendon stump into the palm due to muscle contraction [93,94]. This aspect of the clinical presentation differs significantly from the design of the included studies, which involved transection of the tendon followed by immediate repair under a single general anaesthetic procedure. Retrieval of the tendon stump in human patients risks damage to the tendon tissue [95]. It is reasonable to consider that such damage could compromise healing at the repair site and, therefore, the benefit of MSC therapy. Future studies could involve delayed repair to model the clinical presentation of tendon injury more closely.

### 4.3. Study Limitations

Important limitations to the study findings should be noted. First, the risk of bias arising from the included studies was assessed as being high. The lack of reporting of baseline characteristics resulted in bias arising from the randomization process. Furthermore, given that assessors were not blinded, it was unclear whether this led to bias in measurement of biomechanical properties. Future studies should explicitly report on baseline characteristics and endeavour to use a blinded design.

Heterogeneity in the study designs poses further limitations to generalization of the study findings. Studies varied regarding various aspects of treatment administration, including the MSC source and dosage, method of delivery and the co-administration of growth factors and lubricating biomolecules. Future studies should aim to directly compare variations in each of these parameters to achieve a standardized approach to augmentation of flexor tendon repair using tissue engineering techniques.

Discrepancies between animal and human models of flexor tendon injury mentioned above further obscure potential extrapolation of the study findings to the clinical setting. To our knowledge, no human studies have been conducted on the use of MSCs for the augmentation of flexor tendon repair. Furthermore, due to the limited number of included studies, results concerning rabbit and canine subjects were pooled. This was accounted for during statistical analysis by using the effect size, or standardized mean difference, when comparing control and intervention groups. However, the validity of this study’s conclusions would be improved by collation of results from the same animal species if sufficient studies were available.

## 5. Conclusions

Tendons demonstrate characteristic mechanical properties, which are a function of their hierarchical fascicular structure and ECM components. The purpose of this review was to determine the effect of MSCs on these properties when used to augment surgical repair of transected flexor tendons. The primary clinical complications following flexor tendon repair are adhesions, limiting post-operative range of motion, and rupture due to reduced tensile strength at the repair site. MSCs were able to effectively mitigate adhesion formation, which is likely a result of their immunomodulatory capability. This was not associated with a significant impact on the maximum load. Meta-analyses have shown reduced elastic modulus and increased energy absorption following MSC administration. The clinical significance of these findings is unclear. Reduced adhesion formation would have important consequences for patients’ quality of life by reducing post-operative hand stiffness. However, the ultimate functional benefit is unclear, given evidence that MSCs may alter the viscoelastic behaviour of treated tendons such that energy transfer between muscle and bone is reduced. Limitations of this review are primarily related to the risk of bias of the included papers, study heterogeneity and cautious extrapolation of animal models to the human phenotype of tendon injury and repair. Future studies should justify the animal model used and aim to standardize the therapeutic application of MSCs in the context of flexor tendon repair before their potential application in human trials.

## Figures and Tables

**Figure 1 bioengineering-11-00656-f001:**
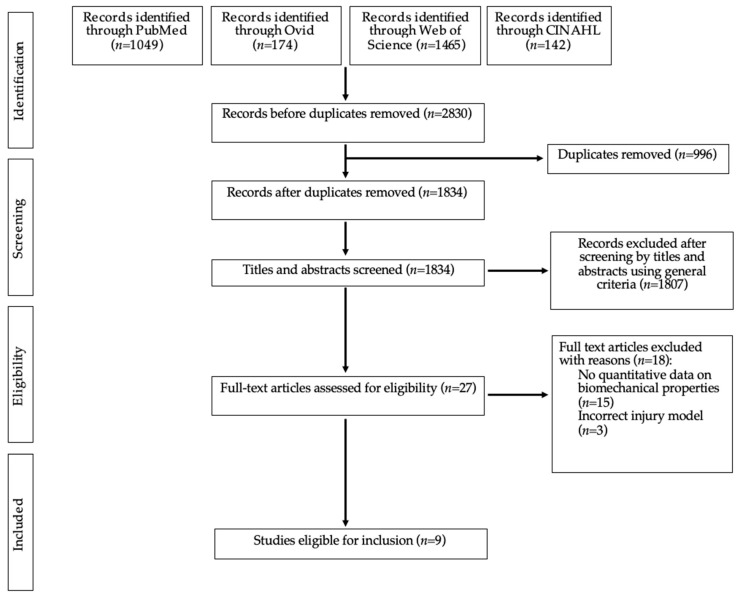
PRISMA flowchart demonstrating article screening and selection.

**Figure 2 bioengineering-11-00656-f002:**
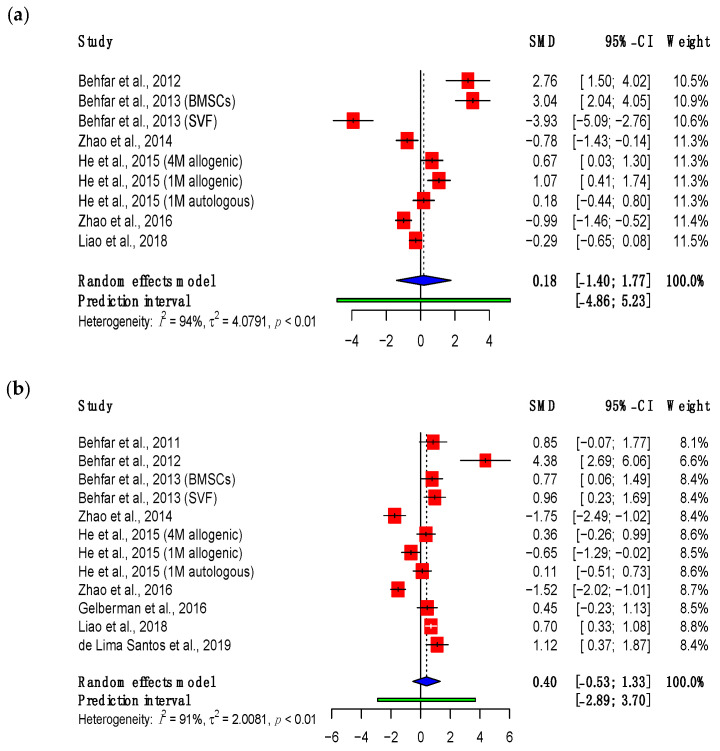
Forest plot showing the maximum load of treated tendons at (**a**) three weeks and (**b**) eight weeks after treatment. (Abbreviations: BMSC, bone marrow-derived mesenchymal stromal cell; CI, confidence interval; M, million; SVF, stromal vascular fraction) [47,48,49,50,51,52,53,54,55].

**Figure 3 bioengineering-11-00656-f003:**
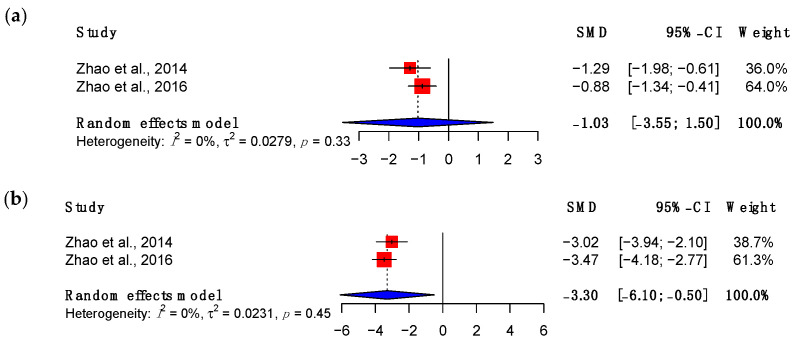
Forest plot showing the friction of treated tendons at (**a**) three weeks and (**b**) six weeks after treatment. (Abbreviations: CI, confidence interval) [50,52].

**Figure 4 bioengineering-11-00656-f004:**
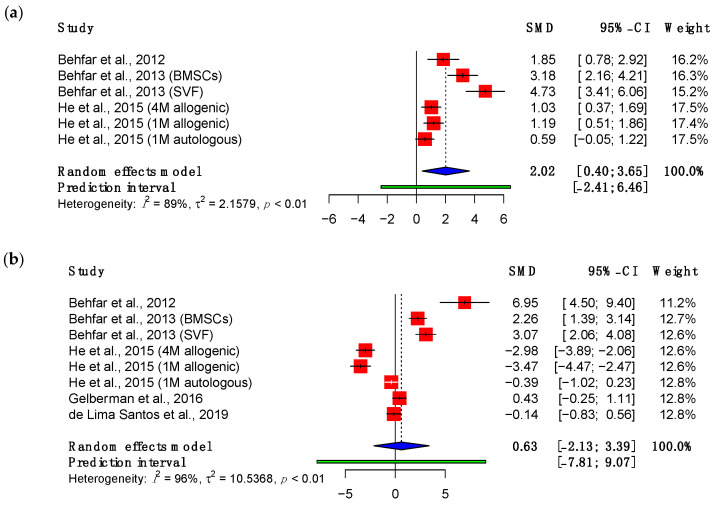
Forest plot showing the maximum stress of treated tendons at (**a**) three weeks and (**b**) eight weeks after treatment (Abbreviations: BMSC, bone marrow-derived mesenchymal stromal cell; CI, confidence interval; M, million; SVF, stromal vascular fraction) [48,49,51,53,55].

**Figure 5 bioengineering-11-00656-f005:**
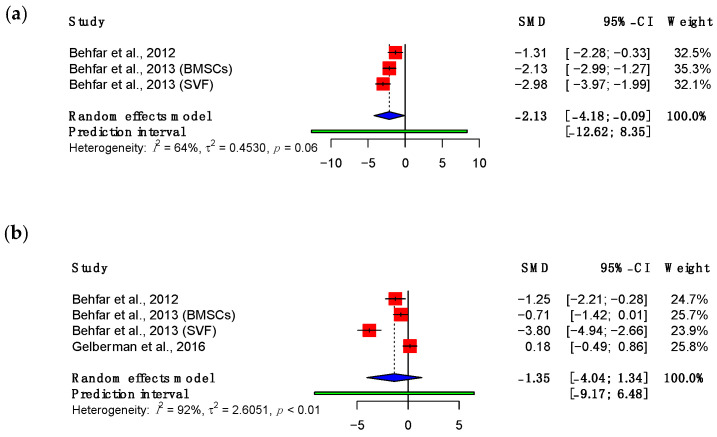
Forest plot showing the maximum strain of treated tendons at (**a**) three weeks and (**b**) eight weeks after treatment. (Abbreviations: BMSC, bone marrow-derived mesenchymal stromal cell; CI, confidence interval; SVF, stromal vascular fraction) [48,49,53].

**Figure 6 bioengineering-11-00656-f006:**
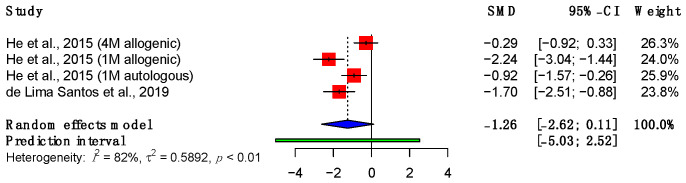
Forest plot showing the Young’s modulus of treated tendons at final follow-up. (Abbreviations: CI, confidence interval; M, million) [51,55].

**Figure 7 bioengineering-11-00656-f007:**
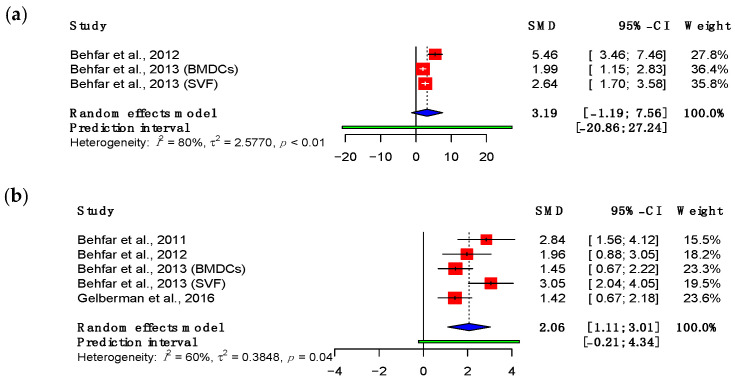
Forest plot showing the energy absorption of treated tendons at (**a**) three weeks and (**b**) eight weeks, or at final follow-up, after treatment. (Abbreviations: BMDC, bone marrow-derived mesenchymal stromal cell; CI, confidence interval; SVF, stromal vascular fraction) [47,48,49,53].

**Figure 8 bioengineering-11-00656-f008:**
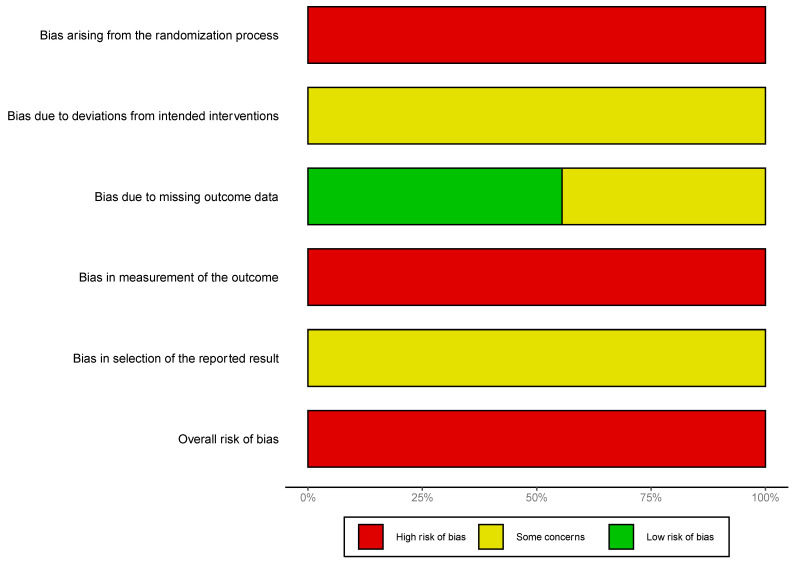
Summary graph showing the overall risk of bias analysis using the RoB 2.0 tool in randomized studies.

**Table 1 bioengineering-11-00656-t001:** PICOS inclusion and exclusion criteria for study selection.

Domain	Inclusion Criteria	Exclusion Criteria
Population	Any animals with completely transected digital flexor tendons.	Studies involving human or cadaveric subjects. Studies investigating repair of other tendons or the healing of tendons which are not completely transected. Ex vivo, in vitro, or in silico studies.
Intervention	Studies investigating allogenic and/or autologous MSC delivery to the injury site in addition to surgical repair. Studies using any cell delivery method, including intratendinous injection, gel droplets and scaffold implants.	Studies involving only cell-free therapies without comparison with MSCs or cell therapies which are not MSCs.
Comparison	Studies that compare the use of MSCs to other tissue engineering techniques or cell-free therapy.	None
Outcome	Studies that provide quantitative outcomes from mechanical testing of flexor tendons after surgery.	Studies which provide only qualitative outcome data.
Study type	Controlled trials, case series, articles published in English with full-text available.	Case reports and review articles.

**Table 2 bioengineering-11-00656-t002:** Study characteristics.

Author, Year	Design	Animal	Cohort Size	Defect Location	Intervention	MSC Source	Cell Delivery Method	Cell Dosage	Repair Method	Post-Operative Weight Bearing	Timing of Sacrifice
Behfar et al., 2011 [47]	RCT	Adult male New Zealand white rabbits, 2.5–3 kg	25 (5 adipose tissue donors, 10 treated, 10 control)	Deep digital flexor tendon, central one third	Fresh stromal vascular fraction from enzymatic digestion of adipose tissue	Allogeneic adipose-derived stromal vascular fraction, obtained from inguinal fat pad	Intratendinous injection in both tendon stumps and the repair site	4 × 10^6^ nucleated cells in 0.2 mL PBS	3-0 monofilament nylon, modified Kessler technique	Immobilization in below-stifle plaster cast for 2 weeks.	8 weeks
					Control: Suture + PBS injection			0.2 mL PBS alone			
Behfar et al., 2012 [48]	RCT	Adult male New Zealand white rabbits, 2.5–3 kg	25 (5 adipose tissue donors, 10 treated, 10 control)	Deep digital flexor tendon, central one third	Allogeneic stromal vascular fraction	Allogeneic adipose-derived stromal vascular fraction, obtained from inguinal fat pad	Intratendinous injection into the suture site	4 × 10^6^ nucleated cells in 0.2 mL PBS	3-0 monofilament nylon, modified Kessler technique	Immobilization in below-stifle plaster cast for 2 weeks.	3 and 8 weeks
					Control: Suture + PBS injection			0.2 mL PBS alone			
Behfar et al., 2013 [49]	RCT	Adult male New Zealand white rabbits	48 (12 donors, 24 treated, 12 control)	Deep digital flexor tendon, central one third	Fresh allogeneic stromal vascular fraction	Allogeneic adipose-derived stromal vascular fraction, obtained from inguinal fat pad	Intratendinous injection at the suture site	4 × 10^6^ nucleated cells of freshly isolated SVF	3/0 monofilament nylon, modified Kessler technique	Immobilization with a below-stifle plaster cast for two weeks.	3 and 8 weeks
					Cultured allogeneic BMSCs	Iliac crest bone marrow from allogeneic donors		4 × 10^6^ cultured BMSCs in 0.2 mL PBS			
					Control: Suture + PBS injection			0.2 mL PBS alone			
Zhao et al., 2014 [50]	RCT	Female mixed-breed dogs, approximately 1 year old, approximately 20 kg	60 dogs, 120 paws	Second and fifth FDP from one forepaw, Zone II-D level	Carbodiimide-derivatized hyaluronic acid, gelatin, and lubricin plus autologous BMSCs stimulated with growth and differentiation factor 5	Tibial bone marrow	“Cell patch” (1 mm-diameter gel droplet composed of collagen/MEM solution MSC and GDF5-5) placed between lacerated tendon ends followed by cd-HA- lubricin surface treatment	8 × 10^5^ (four gel droplets)	4-0 FiberWire1 suture (Arthrex Inc, Naples, FL, USA), modified Pennington technique, reinforced with running suture: 6-0 ProleneTM (Ethicon Inc., Somerville, NJ, USA)	Radial neurectomy was performed after treatment so that dogs could not bear weight and the treated paw was held with a sling in front of the chest for five days; synergistic motion rehabilitation was performed daily from day six until euthanasia.	10 days, 21 days, 42 days
					Suture repair only						
					Normal (uninjured)						
He at al., 2015 [51]	RCT	Female New Zealand White rabbits, 2.5–3 kg	40 rabbits	Rear paws index and ring fingers, FDP, middle of Zone II	Repair + four million allogeneic BMSCs + fibrin glue	Iliac crest bone marrow	Pipetted around the repair site	10^6^ MSCs per tendon	Modified Kessler’s technique	After surgery rabbits were allowed to move liberally.	3 and 8 weeks
					Repair + one million allogeneic BMSCs + fibrin glue						
					Repair + one million autologous BMSCs + fibrin glue						
					Repair + fibrin glue only						
Zhao et al., 2016 [52]	RCT	Mixed-breed dogs	39 dogs, 78 tendons	Second and fifth digit, FDP, Zone II-D level	Repair + cd-HA-lubricin + interpositional graft of 8 × 10^5^ BMSCs and GDF-5	Tibial bone marrow	“Cell patch” (1 mm-diameter gel droplet composed of collagen/MEM solution MSC and GDF5) placed between lacerated tendon ends followed by cd-HA-lubricin surface treatment	8 × 10^5^ cells (four gel droplets)	4-0 FiberWire1 suture (Arthrex Inc., Naples, FL, USA), modified Pennington technique, reinforced with running suture: 6-0 ProleneTM (Ethicon Inc., Somerville, NJ, USA)	Radial neurectomy performed to paralyze the elbow and wrist extensors and prevent weight-bearing. Wrist immobilization in 90° of flexion achieved with a threaded, 1.6 mm diameter K-wire passing from distal radius to the proximal third of the metacarpal bone. Custom jackets immobilized the operated paw in front of the chest. Dogs living after day 21 underwent K-wire removal and started wrist and digit synergistic therapy.	21 and 42 days
					Repair only						
Gelberman et al., 2016 [53]	RCT	Adult mongrel dogs, 20–30 kg	17 dogs, 34 tendons	Second and fifth digits of the right forelimb, FDP, Zone 2	Repair + Heparin/fibrin-based delivery system/nanofiber scaffold + BMP12 + ASC	Subcutaneous adipose tissue	Longitudinally oriented horizontal slits in the centre of each tendon stump followed by insertion of scaffold which was secured with core suture and epitenon suture	7.5 μg BMP12 and 1 × 10^6^ autologous ASCs	Core suture: 8-strand suture of 4-0 multifilament nylon (168; grant) (4-0 Supramid, S. Jackson, Alexandria, Virginia); Epitendinous suture: 6-0 nylon running epitenon suture	Controlled passive motion exercise until euthanasia.	28 days
					Repair + acellular scaffold						
					Repair only						
					Uninjured						
Liao et al., 2018 [54]	RCT	Female New Zealand white rabbits	29 rabbits, 116 tendons	Index and ring digits of the hind paws, FDP, level of proximal phalanx	Scaffold + BMSC	Iliac crest bone marrow	L-lactide and ℇ -caprolactone (PLCL) (Purac Biomaterials, Lincolnshire, IL)—Hyaluronic acid (HA) scaffold	10^5^ MSCs per scaffold	Core suture: modified-Kessler technique, 5/0 prolene (Ethicon, Somerville, NJ). Epitendinous suture: none, due to the small size of the tendons. Scaffolds were wrapped around the repair site and tagged with prolene 6/0 interrupted sutures.	Flexor tendons were divided at the MCPJ to unload the repair. Animals were allowed to move freely without splinting post-operatively.	3 and 8 weeks
					Scaffold						
					Repair only						
de Lima Santos et al., 2019 [55]	RCT	Male New Zealand rabbits, 2–2.5 kg	16 rabbits, 32 tendons	Hind leg, FDS, 1–2 cm from the distal part of the calcaneus	Repair + ASC	Inguinal fat pad	Injection (composition not specified)	1–2 × 10^6^ per injection	Core suture: modified-Kessler technique, Nylon 2/0 (Nylon 2-0; Shalon, Alto da Boa Vista, GO, Brazil). Epitendinous suture: polyglycolic acid 4–0 (Polyglycolic Acid 2-0; Brasuture, Sao Sebastiao da Grama, SP, Brazil)	Free movement without postoperative cast immobilisation.	4 weeks
					Repair only						
					No suture						
					Uninjured						

Abbreviations: ASC, adipose-derived mesenchymal stromal cell; BMSC, bone marrow-derived mesenchymal stromal cell; BMP12, bone morphogenetic protein 12; cd-HA-lubricin, carbodiimide-derivatized gelatin, hyaluronic acid, and lubricin; FDP, flexor digitorum profundus; FDS, flexor digitorum superficialis; GDF5, growth differentiation factor 5; HA, hyaluronic acid; MEM, minimum essential media; MSC, mesenchymal stromal cell; PBS, phosphate-buffered saline; PLCL, L-lactide and ℇ-caprolactone; SVF, stromal vascular fraction.

**Table 3 bioengineering-11-00656-t003:** Results of biomechanical testing.

Author, Year	Intervention	Cohort Size	Max. Load, N	Energy Absorption, N·mm	Max. Stress, N/mm^2^	Max. Strain, %	Elastic Modulus, MPa	Range of Motion/Gliding Resistance/Friction	Significance (If Any)
Behfar et al., 2011 [47]	Stromal vascular fraction	5 (8 weeks)	34.67 ± 3.17	49.12 ± 17.66					*p* < 0.05 for all parameters
	Suture + PBS injection	5 (8 weeks)	8.64 ± 3.85	13.01 ± 3.40					
Behfar et al., 2012 [48]	Stromal vascular fraction	5 (3 weeks); 5 (8 weeks)	13.30 ± 3.98 (3 weeks); 53.10 ± 10.17 (8 weeks)	29.74 ± 3.17 (3 weeks); 96.34 ± 47.84 (8 weeks)	4.43 ± 1.32 (3 weeks); 18.92 ± 1.49 (8 weeks)	12.60 ± 2.04 (3 weeks); 11.01 ± 1.52 (8 weeks)			*p* < 0.05 for maximum load, energy absorption and maximum stress at 3 and 8 weeks
	Suture + PBS injection	5 (3 weeks); 5 (8 weeks)	5.07 ± 1.40 (3 weeks); 14.10 ± 7.44 (8 weeks)	9.07 ± 4.31 (3 weeks); 26.01 ± 8.05 (8 weeks)	2.18 ± 1.10 (3 weeks); 4.7 ± 2.48 (8 weeks)	19.61 ± 7.30 3 weeks); 15.49 ± 4.85 (8 weeks)			
Behfar et al., 2013 [49]	Stromal vascular fraction	6 (3 weeks); 6 (8 weeks)	10 (3 weeks); 35 (8 weeks)	16 (3 weeks); 49 (8 weeks)	20 (3 weeks); 38 (8 weeks)	2 (3 weeks); 1 (8 weeks)			Treatment groups vs. control: *p* < 0.05 for maximum load, energy absorption, and stress at 3 and 8 weeks. SVF vs. BMSC: *p* < 0.05 for energy absorption and stress at 8 weeks.
	BMSCs	6 (3 weeks); 6 (8 weeks)	13 (3 weeks); 34 (8 weeks)	11 (3 weeks); 31 (8 weeks)	13 (3 weeks); 33 (8 weeks)	2 (3 weeks); 2.5 (8 weeks)			
	Suture + PBS injection	6 (3 weeks); 6 (8 weeks)	4 (3 weeks); 27 (8 weeks)	6 (3 weeks); 21 (8 weeks)	6 (3 weeks); 25 (8 weeks)	3 (3 weeks); 3 (8 weeks)			
Zhao et al., 2014 [50]	cd-HA-lubricin + interpositional graft of BMSCs and GDF-5	18 (10 days), 18 (21 days), 16 (42 days)	42 (10 days); 35 (21 days); 44.7 ± 8.5 (42 days)					Work of flexion in N/mm/degree (10 digits per group): 0.28 ± 0.08 (10 days), 0.29 ± 0.19 (21 days), and 0.32 ± 0.22 (42 days) Friction: 0.55 ± 0.15 N (10 days), 0.52 ± 0.2 (21 days); 0.36 ± 0.12 (42 days)	*p* < 0.05 for work of flexion and friction in favour of MSC at 10, 21 and 42 days. *p* < 0.05 for maximum load in favour of suture repair alone at 42 days.
	Suture repair only	16 (10 days), 17 (21 days), 16 (42 days)	38 (10 days); 43 (21 days); 70.2 ± 18.77 (42 days)					0.46 ± 0.19 (10 days), 0.77 ± 0.49 (21 days), 1.17 ± 0.82 (42 days) 0.93 ± 0.3 (10 days), 0.98 ± 0.46 (21 days), 0.62 ± 0.02 (42 days)	
	Normal (uninjured)	10 (0 days)	47 (day 0)					Contralateral, non-operated paw (no incision): approx. 0.2 at all time points (bar-chart estimate) Contralateral, non-operated paw (no incision): approx. 0.05, 0.08, 0.08 (bar-chart estimate)	
He at al., 2015 [51]	four million allogeneic BMSCs + fibrin glue	9 (3 weeks); 9 (8 weeks)	12.5 (3 weeks), 27 (8 weeks)		4.5 (3 weeks), 38 (8 weeks)		60 (3 weeks), 750 (8 weeks)	Post-operative degrees of flexion: 50 (3 weeks), 41 (8 weeks)	*p* < 0.05 for ROM in favour of 4 M allogeneic cells at 3 weeks but not 8 weeks.
	one million allogeneic BMSCs + fibrin glue	11 (3 weeks); 11 (8 weeks)	14 (3 weeks), 19 (8 weeks)		4.5 (3 weeks), 37 (8 weeks)		70 (3 weeks), 500 (8 weeks)	36 (3 weeks), 45 (8 weeks)	
	one million autologous BMSCs + fibrin glue	9 (3 weeks); 11 (8 weeks)	11 (3 weeks); 25 (8 weeks)		3.5 (3 weeks), 48 (8 weeks)		50 (3 weeks), 650 (8 weeks)	38 (3 weeks), 44 (8 weeks)	
	Fibrin glue only	7 (3 weeks); 12 (8 weeks)	10.5 (3 weeks); 24 (8 weeks)		3 (3 weeks); 50 (8 weeks)		40 (3 weeks), 800 (8 weeks)	30 (3 weeks), 46 (8 weeks)	
Zhao et al., 2016 [52]	Repair + cd-HA-lubricin + BMSC + GDF-5	19 (21 days); 20 (42 days)	30 (21 days); 38 (42 days)					Work of flexion in N/mm/degree: 0.25 (21 days); 0.3 (42 days) Friction in N: 0.45 (21 days); 0.5 (42 days)	*p* < 0.05 for work of flexion and gliding resistance in favour of MSC at 21 and 42 days. *p* < 0.05 for failure strength in favour of surgical repair alone at 21 and 42 days.
	Repair	19 (21 days); 20 (42 days)	41 (21 days); 62 (42 days)					0.5 (21 days); 0.9 (42 days) 0.7 (21 days); 0.9 (42 days)	
	Contralateral, non-operated paw cut and sutured immediately post-mortem	8 (0 days)	37 (0 days)						
Gelberman et al., 2016 [53]	Repair + scaffold + BMP12 + ASC	10 (4 weeks)	85	2.6	14	3.3 ± 1.1		PIP + DIP degrees of motion: 35.7	*p* < 0.05 for range of motion in favour of uninjured control.
	Repair + acellular scaffold	15 (4 weeks)	77	2.2	13	3.8 ± 1.4		35.2	
	Repair only	8 (4 weeks)	83	1.7	15	3.1 ± 1.1		41.3	
	Normal (uninjured) tendon from opposite limb	25 (4 weeks)						55	
Liao et al., 2018 [54]	PLCL-HA scaffold + BMSC	15 (3 weeks); 8 (8 weeks)	14 (3 weeks); 28 (8 weeks)					ROM at PIPJ and DIPJ: 43 (3 weeks); 40 (8 weeks)	*p* < 0.05 for maximum load in favour of suture repair alone
	PLCL-HA scaffold	14 (3 weeks); 8 (8 weeks)	15 (3 weeks); 22 (8 weeks)					52 (3 weeks); 48 (8 weeks)	
	Repair only	19 (3 weeks); 8 (8 weeks)	17.5 (3 weeks); 37 (8 weeks)					40 (3 weeks); 48 (8 weeks)	
de Lima Santos et al., 2019 [55]	Suture + ASC	9 (4 weeks)	96.56 (21.27)		11.04 (3.17)		6.25 (2.61)		*p* < 0.001 for all tests relative to uninjured control (ANOVA); *p* < 0.05 for maximum load in favour of ASC.
	Suture alone	10 (4 weeks)	70.82 (24.66)		11.53 (3.88)		12.02 (4.04)		
	No suture	0 (4 weeks)							
	Control (uninjured)	9 (4 weeks)	132.69 (17.48)		44.42 (12.13)		57.80 (33.48)		

Abbreviations: ASC, adipose-derived mesenchymal stromal cell; BMSC, bone marrow-derived mesenchymal stromal cell; BMP12, bone morphogenetic protein 12; cd-HA-lubricin, carbodiimide-derivatized gelatin, hyaluronic acid, and lubricin; N, newtons; mm, millimetre; MPa, megapascal; PLCL-HA, L-lactide and ℇ-caprolactone-hyaluronic acid; PBS, phosphate-buffered saline; SVF, stromal vascular fraction.

**Table 4 bioengineering-11-00656-t004:** Subgroup meta-analyses regarding the MSC source used.

MSC Source	Number of Cohorts	SMD	95% Confidence Interval	p_subgroup_
Maximum load				
3 weeks				
Adipose	2	−0.5882	−43.0557, 41.8792	0.7758
Bone marrow	7	0.3746	−0.8715, 1.6207	
8 weeks				
Adipose	4	1.5782	−1.1612, 4.3176	0.0693
Bone marrow	8	−0.1256	−1.0072, 0.7561	
Stress				
3 weeks				
Adipose	2	3.2620	−15.0288, 21.5528	0.2390
Bone marrow	4	1.4436	−0.3370, 3.2242	
8 weeks				
Adipose	4	2.4274	−2.5892, 7.4439	0.0831
Bone marrow	4	−1.1350	−5.3330, 3.0629	

## Data Availability

The data are contained within the article and Appendix A.

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
