# Peer review of "Mesenchymal Stromal Cells for the Enhancement of Surgical Flexor Tendon Repair in Animal Models: A Systematic Review and Meta-Analysis"

_bioengineering, 2024, doi:10.3390/bioengineering11070656_

Round 1
Reviewer 1 Report
Comments and Suggestions for Authors
Thank you for submitting your manuscript entitled "Augmentation of Flexor Tendon Repair with Mesenchymal Stem Cells: A Systematic Review and Meta-Analysis" for consideration in our journal. After careful review, I have identified some areas where some revisions are needed to strengthen the manuscript before it can be accepted for publication.
There are instances throughout the manuscript where clarity and transparency could be improved. For example, the roles of "AE and AE" in the screening process need to be clearly defined. Providing clear explanations and context will enhance the readability and understanding of the manuscript.
Incomplete Reporting: Certain essential details, such as baseline characteristics of the included studies and criteria for assessing the risk of bias, are missing. Including these details is crucial for evaluating the validity and reliability of the study's findings. Please ensure that all relevant information is provided to facilitate a comprehensive understanding of the research methodology.
Statistical Analysis and Interpretation: The statistical analysis and interpretation need to be more comprehensive. Key details, such as significance levels, confidence intervals, and p-values, should be included to convey the strength of the evidence and the reliability of the findings accurately. Enhancing the statistical reporting will improve the rigor and credibility of the study.
The discussion and conclusion sections require revision to provide a more thorough analysis of the findings and their implications. It is important to critically appraise the results, address potential limitations, and suggest avenues for future research. A concise summary of the key findings and their clinical relevance should be provided in the conclusion.
I recommend revising the methodology to adhere to established guidelines and ensuring consistency in reporting across all sections of the manuscript. This will enhance the credibility and impact of the research.
Overall, I believe that addressing these revisions will significantly strengthen the manuscript and make it suitable for publication in our journal.
Comments on the Quality of English LanguageThe English language used in the mentioned article is generally clear and understandable. However, there are areas where improvements can be made for better clarity and coherence.
-
Some sentences are lengthy and complex, which can make them difficult to follow. Simplifying these sentences or breaking them down into smaller, more digestible segments would improve readability.
- While the article appears to have been proofread, there are still instances of typographical errors, missing words, or grammatical inconsistencies that need to be addressed.
Overall, the English language in the article is adequate, but implementing these suggestions will help enhance clarity, coherence, and overall readability.
Author Response
Dear Reviewer,
Thank you very much for taking the time to review our work and provide your suggestions for improvements. We have taken your comments into account and believe these have considerably improved our manuscript.
In response to your comments, we have altered our manuscript as follows:
- We have expanded on each section of the Materials and Methods to clarify the processes involved, including the screening process, data extraction and analysis. The roles of each author are described in more detail. For clarity we have added the second letter of the surnames for the second and third authors.
- We have re-reviewed the included studies to ascertain further detail regarding baseline characteristics of the animal subjects. Additional details which have now been added are highlighted within Table 2. Further detail could not be provided due to limited reporting within the included studies themselves. We have highlighted this as a contributor to the high risk of bias in the Results and Discussion. A new Limitations section has now been included within the Discussion, which mentions this limitation of the included studies.
The criteria for assessing risk of bias have now been described more comprehensively within the Materials and Methods. Individual risk of bias assessments for each included paper have now been included within the supplementary materials. - We have added the p-values for each of the meta-analyses within the Results section, along with the standardized mean difference (SMD) and confidence intervals. The SMD and confidence intervals are also presented within the forest plots.
- We have now expanded on the Discussion and Conclusion to provide a more thorough analysis. A summary of the main study findings and their interpretation has been provided at the beginning of the Discussion. The Discussion has been re-organized to include a section on the challenges of tissue engineering in tendon repair and consequent considerations for future research. Further critical appraisal is included in the added section regarding the limitations of this review’s findings. The Conclusion has been expanded to clarify the potential clinical implications of the review’s findings.
- We have used the PRISMA checklist to guide revisions to the methodology, so that the review now aligns with the checklist more closely. The PRISMA checklist is attached in the supplementary files. Checklist items now addressed in more detail have been highlighted within the attached PRISMA checklist.
- We have reworded several sentences throughout the text to improve readability. These are highlighted using track changes.
The article has been proofread further to address the errors mentioned.
Reviewer 2 Report
Comments and Suggestions for Authors
In an attempt to conduct a meta-analysis on the effects of MSC on tendon rupture healing, the authors mixed results from two very different species. The term "animal" cannot combine rabbits and Dogs, so why were human studies excluded?
The reviewer suggest that these two species should be analysed separately and also human studies if any should be included and analysed separetely.
Author Response
Dear Reviewer,
Thank you very much for taking the time to review our manuscript and providing your suggestions for improvements.
Human studies have not been included because, to our knowledge, only pre-clinical trials regarding the use of MSC therapy for tendon repair have been performed. The method of statistical analysis used accounts for variability between species by using the standardized mean difference (SMD), or effect size, to compare outcomes of different studies, rather than using the raw data. The rationale for this analytic method has been specifically mentioned within the Materials and Methods. In addition, no differences were found in the overall trends of studies investigating either rabbits or dogs. However, we have now included a paragraph on the potential limitations of animal models and their resemblance to humans in the Discussion.
We are very grateful for your comments and believe they have improved our manuscript greatly.